# Controlled Nanostructuration of Cobalt Oxyhydroxide Electrode Material for Hybrid Supercapacitors

**DOI:** 10.3390/ma14092325

**Published:** 2021-04-29

**Authors:** Ronan Invernizzi, Liliane Guerlou-Demourgues, François Weill, Alexia Lemoine, Marie-Anne Dourges, Isabelle Baraille, Delphine Flahaut, Jacob Olchowka

**Affiliations:** 1CNRS, University of Bordeaux, Bordeaux INP, ICMCB UMR CNRS #5026, F-33600 Pessac, France; Ronan.INVERNIZZI@icmcb.cnrs.fr (R.I.); Liliane.Guerlou-Demourgues@enscbp.fr (L.G.-D.); Francois.Weill@icmcb.cnrs.fr (F.W.); 2RS2E, Réseau Français sur le Stockage Electrochimique de l’Energie, FR CNRS #3459, CEDEX 1, F-80039 Amiens, France; dflahaut@univ-pau.fr; 3ALISTORE-ERI European Research Institute, FR CNRS #3104, CEDEX 1, F-80039 Amiens, France; 4CNRS/University of Pau and Pays de l’Adour/E2S UPPA, Institut des Sciences Analytiques et de Physicochimie pour l’Environnement et les Matériaux—UMR 5254, F-64000 Pau, France; alexia.lemoine@univ-pau.fr (A.L.); isabelle.baraille@univ-pau.fr (I.B.); 5Institut des Sciences Molaires, University of Bordeaux, UMR 5255, F-33405 Talence, France; marie-anne.dourges@u-bordeaux.fr

**Keywords:** nanostructuration, supercapacitors, cobalt oxyhydroxide, ionic liquids, energy storage, surface modification, nanomaterial

## Abstract

Nanostructuration is one of the most promising strategies to develop performant electrode materials for energy storage devices, such as hybrid supercapacitors. In this work, we studied the influence of precipitation medium and the use of a series of 1-alkyl-3-methylimidazolium bromide ionic liquids for the nanostructuration of β(III) cobalt oxyhydroxides. Then, the effect of the nanostructuration and the impact of the different ionic liquids used during synthesis were investigated in terms of energy storage performances. First, we demonstrated that forward precipitation, in a cobalt-rich medium, leads to smaller particles with higher specific surface areas (SSA) and an enhanced mesoporosity. Introduction of ionic liquids (ILs) in the precipitation medium further strongly increased the specific surface area and the mesoporosity to achieve well-nanostructured materials with a very high SSA of 265 m^2^/g and porosity of 0.43 cm^3^/g. Additionally, we showed that ILs used as surfactant and template also functionalize the nanomaterial surface, leading to a beneficial synergy between the highly ionic conductive IL and the cobalt oxyhydroxide, which lowers the resistance charge transfer and improves the specific capacity. The nature of the ionic liquid had an important influence on the final electrochemical properties and the best performances were reached with the ionic liquid containing the longest alkyl chain.

## 1. Introduction

Nanomaterials are generating an intense research interest in the energy storage area due to their numerous advantages compared to their bulk counterparts [1,2]. Indeed, they offer greatly improved ionic diffusion and electronic conductivity compared to micron-sized particles, which allows faster charge/discharge kinetics [1]. Additionally, nanoparticles’ size permits them to better accommodate the stress induced by repeated charge/discharge cycles and thus, improves the lifetime of the electrode material [3]. These characteristics make them ideal candidates for electrode materials in hybrid supercapacitor applications, where the high power density and the long cycle life provided by nanostructuration are associated with the high capacity generated by redox reactions [4].

Among the different nano-oxides/(oxy)hydroxides tested as electrode materials, non-stoichiometric cobalt oxyhydroxide (β(III)-H_x_CoO_2_) has attracted a great deal of attention due to its excellent electronic conductivity, provided by the presence of tetravalent cobalt in its structure and its good ionic diffusion due to its layered structure [5]. Furthermore, the potential use of cobalt oxyhydroxide goes beyond the electrochemical energy storage domain since it is an object of intense study as an OER (Oxygen Evolution Reaction) catalyst or for photocatalysis purposes [6,7,8,9,10]. In all these applications, a high specific surface area is required to optimize the electrode/electrolyte interface and/or the reactivity of the electrode material as the overall performances strongly depend on surface reactions. Several synthesis approaches have been reported in the literature, leading to nanostructured cobalt oxyhydroxides with various morphologies, such as nanoflakes, nanorods, nanosheets, nanodiscs, thin films, or even as single crystals [11,12,13,14,15,16,17,18,19,20]. Most of the time, the nanostructuration strategy consists of adding an electronic conductive carbon source or to grow cobalt oxyhydroxide on a (porous) support; however, few works have investigated the synthesis parameters that influence nanostructuration [21,22]. Ionothermal synthesis is a perfectly adapted approach to develop nanostructured polycrystalline materials [23,24]. Ionic liquids (ILs) can be used as a solvent, capping agent, template, or even as reactants for the synthesis of inorganic or hybrid nanoparticles (NPs) with controlled morphologies [25,26,27,28,29,30]. In addition to the fact that ILs allow controlling the nanoparticles morphology and aggregation, they can also functionalize nanoparticle surfaces during synthesis, and thus, modify or bring new properties [31,32,33]. This surface modification is reported to be beneficial for electrode materials due to a synergetic effect between highly ionic conductive IL and inorganic materials that permits easier hydrogen adsorption/desorption processes and enhances transport properties [32,34,35,36]. However, the influence of the chemical nature of ionic liquids is poorly studied and remains unclear. In this work, we explore the influence of the precipitation medium and the effects of a series of 1-alkyl-3-methylimidazolium bromide ionic liquids on the crystallinity, morphology, porosity, and chemical composition of β(III) cobalt oxyhydroxide. Then, the effects of nanostructuration on energy storage performances are investigated in KOH aqueous electrolytes.

## 2. Experimental Section

### 2.1. Synthesis

Method 1: β(III) cobalt oxyhydroxide was prepared by reverse coprecipitation. First, 3.18 g of Co(NO_3_)_2_.6H_2_O were dissolved in 300 mL of distilled water. Then, the nitrate solution was slowly added to 11 mL of 2 M NaOH under stirring. A blue precipitate (α-Co(OH)_2_) appeared and quickly turned pink (β(II)-Co(OH)_2_) and then brown (partially oxidized cobalt hydroxide). In order to speed up the oxidation process of the freshly precipitated cobalt hydroxide, 7.5 mL (3 times the required volume) of NaClO (4 M) were added dropwise to the solution which turned black. Following this, the solution was stirred for 36 h at room temperature, centrifuged several times at 4000 rpm until the supernatant exhibited a neutral pH, and was then dried at 45 °C overnight.

Method 2: a direct coprecipitation approach was used where 2M NaOH solution was slowly added (drop-by-drop over a few minutes) into a nitrate solution before the same oxidation, centrifugation and drying steps in method 1 were used.

The same process was also used for the synthesis of cobalt oxyhydroxide functionalized with ILs, except that 1-alkyl-3-methylimidazolium bromide (alkyl = ethyl (2C), hexyl (6C) and decyl (10C)) ionic liquids are added to the starting nitrate solution with a molar ratio Co/IL of 1.

Pure β(III) cobalt oxyhydroxide prepared using method 1 and method 2 were designated as β3-pH_14_ and β3-pH_↗_ respectively. Indeed, the starting precipitation medium for method 1 was at pH ≈ 14 whereas it is slightly acidic for method 2 and increased upon adding NaOH. The IL functionalized materials were named β3-pH_↗_(IL 2C), β3-pH_↗_(IL 6C) and β3-pH_↗_(IL 10C), depending on the C number, which was the length of alkyl chain present on the 1-alkyl-3-methylimidazolium bromide ionic liquid.

### 2.2. Powder X-ray Diffraction

XRD patterns were recorded on a Philips Panalytical X’Pert Pro diffractometer (Panalytical, Almelo, The Netherlands), equipped with a cobalt source; K_α_ radiation (λ_Kα1_ = 1.789 Å and λ_Kα2_ = 1.793 Å) was used in order to avoid fluorescence in the sample, which is problematic with traditional copper sources. The powder diffraction patterns were recorded for about 15 h in the 10–110° (2*θ*) angular range, with a 0.02° (2*θ*) step size and a 2.022° (2*θ*) active width in the detector.

### 2.3. Inductively Coupled Plasma—Optical Emission Spectroscopy (ICP-OES)

Co and Na elements were quantified in our samples using a Varian 720ES ICP-OES spectrometer (Varian, Palo Alto, CA, USA). The samples were prepared by dissolving ~25 mg of powder in HCl and the solutions were heated until full dissolution of powders was achieved. Then, the solutions were diluted in order to obtain a concentration of the target element between 1 and 200 mg/L. The solution was then introduced in a nebulization chamber along with an argon flow to create an aerosol. Five measurements were realized for each sample for accuracy.

### 2.4. CHNS Analysis

To quantify the percent of hydrogen in a sample, 1.5 mg of sample was burned at 920 °C in tin foil in an excess of oxygen. The combustion products were collected (H_2_O) and quantified using a Thermo Flash EA 111E Series apparatus (Thermo, Waltham, MA, USA). Two measurements were conducted for each sample to control the accuracy.

### 2.5. Iodometric Titration

Mean oxidation states of Co were determined by iodometric titration.

Thirty milligrams of powder were dissolved in 5 mL of distilled water, 10 mL of KI solution at 10 g/L, and 5 mL of 12M HCl solution (37%). The solution was heated until total dissolution of the powder occurred (yellow color). Immediately after, the solution was titrated using sodium thiosulfate in order to obtain *Veq,* which represented the volume of added sodium thiosulfate when the yellow solution turned pink–violet.

The I^−^ ions contained in KI were oxidized to I_2_ and reduced to the divalent state all the transition metals that had a mean oxidation state superior to 2, according to the following redox reaction:$$\left(x-2\right)I^{-}+{\mathrm{Co}}^{x+}=>{\mathrm{Co}}^{2+}+\frac{x-2}{2}I_{2}$$

The I_2_ molecules were then titrated using sodium thiosulfate (Na_2_S_2_O_3_, 0.1 M):$$2\;{\left(S_{2}O_{3}\right)}^{2-}+I_{2}\to \;S_{4}O_{6}{}^{2-}+2I^{-}$$

To calculate the mean oxidation state of cobalt, the mole number of the formed I_2_ can be expressed as:$$n_{I_{2}}=\frac{x-2}{2}n{\mathrm{Co}}^{x+}=\frac{1}{2}\left[S_{2}O_{3}{}^{2-}\right]*Veq$$
where *Veq* is the volume of sodium thiosulfate added at equivalence. It is therefore possible to determine *x*:$$x=\frac{\left[S_{2}O_{3}{}^{2-}\right]*Veq}{n{\mathrm{Co}}^{x+}}+2$$

The following methodology was used to establish a chemical formula of materials, such as H_w_^+^Na_y_^+^Co^x+^O_2_·n(H_2_O); once the concentrations of Na and Co (ICP-OES) and H (CHNS) were measured, the molar ratio of each element to Co was calculated. The proton amount (w) was deduced by taking the total electroneutrality into account, and the amount of water molecules was reached by subtracting the proton amount from the total H element amount.

### 2.6. Thermogravimetric Analyses

Thermogravimetric analyses were performed on a Setaram TGA 92 (Setaram, Lyon, France). The experiments were carried out under air at a heating rate of 5 °C·min^−1^.

### 2.7. Electric Conductivity Measurements

Electric conductivity measurements were performed using the four-probe technique, using direct current in the 230−400 K temperature range. Because of their instability beyond 400 K, the materials could not be sintered and the measurements were performed on pellets (10 mm in diameter) obtained by compacting ~250 mg of powder at 5 tons.

### 2.8. Scanning Electron Microscopy

The morphology of cobalt oxyhydroxide/IL nanohybrid samples were investigated on a HITACHI 4500-S (Hitachi, Tokyo, Japan).

### 2.9. Transmission Electron Microscopy

Images of the primary particles and electron diffraction patterns were obtained using a JEOL 2100 transmission electron microscope (JEOL, Tokyo, Japan). Prior to observation, the samples were deposited on a formavar/carbon supported film on a copper grid.

### 2.10. Specific Surface Area and Porosity

Surface area and pore structure were explored by recording nitrogen adsorption isotherms at 77 K with micromeritics 3Flex equipment (Micromeritics Corp., Norcross, GA, USA). Before analysis, the samples were degassed at 60 °C under vacuum for 15 h to reach a pressure less than 10 µm Hg.

The specific surface area was obtained using the BET equation applied between 0.01 and 0.25 relative pressure (p/p_0_) [37].

Calculations were performed using the density functional theory (DFT) with the micromeritics software package to determine the pore size distribution. This method took into account the fact that two phenomenon capillary condensation and multimolecular adsorption can provide pore size analysis over a micropore and mesopore range with a good accuracy [38,39].

### 2.11. X-ray Photoelectron Spectroscopy

XPS analyses were performed on a Thermo K-alpha spectrometer (Thermo, Waltham, MA, USA) using focused monochromatized Al Kα radiation (hν = 1486.6 eV). The analysis area was an ellipse of 200 × 400 µm^2^. Cu double tape was used to fix the powder samples to the sample holder. The core peaks were recorded with a pass energy of 20 eV and an energy step of 0.1 eV. A neutralizer gun was needed for charge compensation. No evidence of sample beam damage was observed in the acquired spectra. XPS spectra processing and quantification were obtained using CASA XPS software. The baseline was a Shirley-type background. Surface composition was based on Scofield’s relative sensitivity factors [40].

### 2.12. Electrode Preparation

The electrode was prepared with a mixture of active material/carbon black/polytetrafluoroethylene at a weight ratio of 80/15/5. The final working electrode was pressed at 5 bars on nickel foam for one minute. Each electrode contained a mass loading of 5 mg/cm^2^. Electrochemical measurements were carried out in a basic solution of 5 M KOH in a 3-electrode mode at 25 °C. An Hg/HgO electrode was used as the reference electrode and a platinum wire was used as the counter electrode. EIS measurements were performed under open-circuit conditions. The working electrode, the reference electrode and the counter electrode were placed in the electrolyte in a way to form an equilateral triangle. Each electrode was separated from the others by a distance of 16 mm. A small AC perturbation amplitude of 10 mV versus the open-circuit potential was applied in a frequency range from 100 kHz down to 0.1 Hz.

## 3. Results and Discussion

The XRD patterns of the prepared compounds are shown in Figure 1 for comparison purposes. The XRD pattern corresponding to β3-pH_14_ (black curve, prepared by reverse precipitation according to method 1) can be fully indexed using a rhombohedral cell in the space group *R-3m* (*a* = 2.84 Å and *c* = 13.14 Å) and confirms the presence of a pure β3 cobalt oxyhydroxide phase [5]. The interlamellar distance in this layered phase corresponds to *c*/3 (4.38 Å) and cell parameter *a* corresponds to the metal–metal distance in the slab. XRD patterns of all compounds obtained by direct co-precipitation (method 2) are similar to β3-pH_14_ in terms of their general shapes, but reveal broader reflection peaks suggesting a lower crystallinity. The use of the Scherrer equation on the (*003*) reflection peaks, which allows to calculate the thickness of the coherent domains in the direction of slab stacking, reveals coherent domains around 10 nm and 3 nm, for β3-pH_14_ and β3-pH_↗_ respectively. Moreover, the drop in the intensity ratio between the (*003*) and (*110*) reflections notes a change in the crystallite morphology between β3-pH_14_ and β3-pH_↗_. Introducing ILs during synthesis does not affect the structure since the diffraction patterns look similar to those without IL (β3-pH_↗_) with no supplementary reflection peak detected. However, the broadening of the (*003*) reflection for β3-pH_↗_(IL 10C) allows to consider that the presence of IL with the longest alkyl chain significantly decreases the thickness of the coherent domains (1.5 nm versus 3 nm for β3-pH_↗_), that is the number of stacked slabs, whereas this is not the case for β3-pH_↗_(IL 2C) and β3-pH_↗_(IL 6C) (Table 1).

SEM and TEM images, gathered in Figure 2 and Appendix A, support the results observed by X-ray diffraction. Figure 2a,b shows that β3-pH_14_ is characterized by well-defined hexagonal platelets between 60 and 100 nm in length that are randomly aggregated. In addition, SEM (Figure 2a) also suggests the presence of macroporous cavities/spaces inside the aggregates of platelets. On the other hand, precipitation in the cobalt rich medium (method 2) leads to smaller and less defined particles, as illustrated Figure 2c–e. The particles seem more densely agglomerated, as shown in Figure 2c, and thus, lead to more compact aggregates. TEM images of β3-pH_↗_ (Figure 2d,e) show that the platelet-like aggregates have a length of around 20–30 nm and are composed of several nano-crystallites (Figure 2e), which is in perfect agreement with the small coherent domains observed by X-ray diffraction (Table 1). Electron diffraction performed on a platelet validates its polycrystalline conformation, as illustrated by the circles attributed to (*102*), (*104*) and (*110*) reflections (Figure 2f).

At first glance, the presence of ionic liquids during synthesis does not seem to significantly affect the primary particles nor their agglomerations (Figure 3a–c). Indeed, SEM images of β3-pH_↗_(IL 2C), β3-pH_↗_(IL 6C) and β3-pH_↗_(IL 10C) reveal similar dense aggregates to those of β3-pH_↗_. However, with a higher magnification, the presence of IL creates the impression that the pseudoplatelets observed for β3-pH_↗_ could not be distinguished anymore, as shown for β3-pH_↗_(IL 10C) in Figure 3d,e, which may imply a change in the crystallites agglomeration. Nevertheless, as for the material synthesized without ionic liquids, small crystallites of around 5 nanometers, confirmed as β(III)-H_x_CoO_2_ by electron diffraction, can be clearly distinguished and are consistent with the broad reflections observed by X-ray diffraction (Figure 3e,f and Table 1).

To further observe the influence of the synthesis conditions and the nature of the ionic liquids, specific surface area (SSA) and pore size distribution measurements were performed. Figure 4a shows the N_2_ adsorption/desorption isotherms and the corresponding SSA are reported in Table 1. All cobalt oxyhydroxide samples exhibit a type IV isotherm revealing the presence of micropores and mesopores in materials obtained by co-precipitation (method 2), whereas mostly mesopores can be expected for β3-pH_14_ prepared by reverse co-precipitation. The specific surface area of β3-pH_↗_ (160 m^2^/g) increases more than 100% compared to β3-pH_14_ (77 m^2^/g), which perfectly concurs with the smaller particle sizes observed by microscopy. Addition of ionic liquids during the synthesis further strongly increases the SSA (Table 1). The effect of the length of alkyl chain on imidazolium seems to have a quite limited effect since β3-pH_↗_(IL 2C), β3-pH_↗_(IL 6C) and β3-pH_↗_(IL 10C) have rather comparable SSA values (256, 246 and 232 m^2^/g respectively), even if the general tendency is that the shorter the alkyl chain on imidazolium ring, the higher the SSA. Such elevated SSA values for polycrystalline cobalt oxyhydroxide are among the highest ever reported in the literature, along with those of Wen et al. (241 m^2^/g) [8,12,15,21,41,42,43]. Similarly, porosity is greatly influenced by the experimental conditions, as illustrated in Figure 4b–d, which compares the pore size distribution between all materials. Figure 4b shows that β3-pH_14_ essentially contains large mesopores and macropores, which perfectly agrees with the cavities observed using SEM (Figure 2a), whereas β3-pH_↗_ is composed in its majority of a population of small mesopores centered on ~6 nm in diameter, in perfect agreement with the compact aggregates found by SEM (Figure 2c). Therefore, β3-pH_14_ presents an overall porosity higher than β3-pH_↗_ due to its large mesopores/macropores; however, its porosity becomes much lower when only “small” pores with a diameter of less than 20 nm are considered (0.9 and 0.16 cm^3^/g for β3-pH_14_ and β3-pH_↗_, respectively). The presence of ionic liquids during the material preparation induces pore sizes slightly larger than for β3-pH_↗_ (Figure 4c) and leads to a tremendous increase the in porosity (Figure 4d and Table 1). Once again, the effect of the length of alkyl chain is not clear; β3-pH_↗_(IL 2C) and β3-pH_↗_(IL 10C) have a similar porosities (0.43 cm^3^/g), higher than that of β3-pH_↗_(IL 6C) (0.31 cm^3^/g). Thus, it seems that the ionic liquid with the intermediate alkyl chain affects the nanostructuration less (or in another way) than the two others.

These results clearly demonstrate the impact of synthesis conditions (forward versus reverse co-precipitation) and of the use of ionic liquids as surfactant and template for mastering nanostructuration. Reverse co-precipitation in a strong basic medium (β3-pH_14_) favors crystal growth and thus leads to bigger crystallites compared to forward precipitation where a high cobalt concentration (supersaturation) favors nucleation and thus, small crystallites, for β3-pH_↗_ Therefore, the nature of the precipitation medium is the first experimental key parameter for controlling nanostructuration. Then, the use of surfactant/template, ionic liquids in this work, is the second key parameter to further increase SSA and the mesoporosity in order to develop nanostructured materials. Indeed, it is well known that ionic liquids can adsorb on a nanoparticle’s surface to prevent it from growing, to stabilize it and/or to modify its agglomeration to generate porous materials, as demonstrated in this work by the extended SSA and enlarged mesoporosity [27,44,45]. Moreover, ILs can also functionalize a nanoparticle (NP) surface by creating stable NP–IL bonds, thus leading to hybrid nanomaterials [31,45].

The presence of ionic liquid tethered to the final cobalt oxyhydroxides was detected by X-ray photoelectron spectroscopy and confirmed by infrared analyses (Appendix A). Indeed, the β3-pH_14_ and β3-pH_↗_ survey spectra (Figure 5) presented signatures of cobalt, oxygen and sodium elements (with Co 2p, O 1s and Na 1s core peaks) assigned to β3 cobalt oxyhydroxide [46]. C 1s core peak is related to adventitious carbon and Cl 2p to synthesis residue. The appearing of N 1s core peaks in pH_↗_(IL 2C), β3-pH_↗_(IL 6C) and β3-pH_↗_(IL 10C) survey spectra suggest the surface composition changed with functionalization of ILs. On the other hand, the bromide anion was only detected for IL 10C and was located at 181.8 eV for the Br 3p_3/2_ peak [47]. Similarly, characteristic vibration bands of IL 10C were also detected by IR spectroscopy for β3-pH_↗_(IL 10C) (Appendix A). Nonetheless, an investigation of the high-resolution core peaks is needed to validate IL functionalization.

The N 1s core peaks are shown in Figure 6 for the three functionalized samples. Three nitrogen environments were identified according to the alkyl chain length. N 1s peak components were set at 399.6 eV for ammonium, 401.1 ± 0.4 eV for the two sp^2^ nitrogen atoms in the imidazolium ring and 407 eV for nitrates [48,49,50,51]. The N/Co atomic percentage ratio allows evaluating functionalization, depending on the length of the alkyl chain. We obtained 0.003, 0.01 and 0.02, respectively, for IL 2C, IL 6C, and IL 10C, indicating that the IL functionalization of β3-CoOOH increased with alkyl chain length and presented values coherent with surface modification.

The carbon core peaks (Appendix A) can be decomposed into six components, located at 285 eV (adventitious carbon and aliphatic chain), 286.2 eV (C–O and aliphatic C–N) and 287.1 eV (aromatic C–C*–N and the N–C*–N in the imidazolium ring), 288.6 eV (–C=O), 289.3 eV (–CO_2_) and shake-up satellites around 291.7 eV [50,51].

Co 2p core peaks of bare and functionalized H_x_CoO_2_ materials are displayed in Appendix A. No significant change is observed after IL functionalization. The Co 2p spectra exhibit two peaks, Co 2p_3/2_ (B.E. = 780.3 eV) and Co 2p_1/2_ (B.E. = 795.0 eV) according to the spin-orbit coupling associated with two satellites located at 790.3 eV and 805 eV. Based on the literature, these characteristics are assigned to cobalt, mainly in the +3 oxidation state [52,53]. The oxygen 1s and Na peaks are discussed in the Appendix A.

As previously mentioned, β(III) cobalt oxyhydroxide is a non-stoichiometric cobalt oxyhydroxide. Thus, to gain deeper insights into the influence of synthesis conditions, elementary analyses coupled to iodometric titrations were performed to determine the chemical composition of the prepared materials. Considering the CoO_2_ layers, the negative charge (−4) provided by the two oxygen atoms is compensated by the positive charge of Co^n+^, Na^+^ and H^+^. Once the amount of Na and Co established by ICP-OES and the mean oxidation state of cobalt was determined by iodometric titration, the amount of H^+^ was deduced to compensate for the charge. Then, the remaining hydrogen atoms detected by CHNS were attributed to water molecules that can be structural water (inside the interlayer spacing) or adsorbed on the surface. For materials synthesized in ionic liquid medium, the presence of ILs does not allow to determine an accurate composition in water/ILs. The final formulae can be written as H_v_^+^Na_w_^+^(H_2_O)_y_Co^z+^O_2_(IL)_δ_ and are reported in Table 2. The amount of sodium ranges from 1.4 to 3.2 wt.% for the different materials and the mean oxidation state of cobalt was from 3.05 to 3.20. It can be seen that the presence of ionic liquids, which possess a reducing character due to the imidazolium cation, tends to slightly diminish the mean oxidation state of cobalt, whereas it remains similar for β3-pH_↗_ (3.2) and β3-pH_14_ (3.15) (see Table 2). In parallel, the lower weight ratio of cobalt in the materials synthesized in presence of ionic liquid is most probably due to a higher amount of water being adsorbed because of a higher SSA (see TGA, Appendix A) and, to a lesser extent, to the presence of ILs in the final materials. A decrease in the cobalt mean oxidation state for the β3-pH_↗_(IL) series, conjugated with the presence of ionic liquid, strongly impact the electronic conductivity. Indeed, the room temperature electronic conductivity drops four orders of magnitude between the material with and without ILs (from 2.5 to 2 × 10^−4^ S·cm^−1^ for β3-pH_↗_ and β3-pH_↗_(IL 10C), respectively, see Figure 6d). Nevertheless, it can be highlighted that the electronic conductivity of β3-pH_↗_ is very well maintained and becomes even higher than that of β3-pH_14_, even though the coherent domains size decreases. Finally, modifications of the synthesis conditions only lead to minor changes in the chemical composition of non-stoichiometric β(III) cobalt oxyhydroxide, especially considering the limited accuracy of the iodometric titration, however they have a strong impact on the morphology, porosity and electronic conductivity.

### Electrochemistry

The impact of nanostructuration on the electrochemical energy storage properties was evaluated using cyclic voltammetry (CV), galvanostatic charge/discharge (GCD) and electrochemical impedance spectroscopy (EIS) on electrodes with a mass loading of ~5 mg/cm^2^ [54]. The electrochemical behaviors of all the obtained β(III) phases are illustrated in Figure 7. Voltammograms of β3-pH_↗_ and β3-pH_14_ recorded at 5 mV/s in 5M KOH can be compared in Figure 7a. Both electrode materials exhibit a battery-like profile with reversible oxidation peaks situated at 0.1 V and ~0.43 V versus Hg/HgO corresponding to the redox couple Co^4+^/Co^3+^ [55,56]. The peaks at 0.1 V and 0.43 V can be attributed to the transformation of β3-H_x_CoO_2_ to a β’3-H_x’_CoO_2_ phase and β’3-H_x’_CoO_2_ to a new β’’3-H_x’’_CoO_2_ phase, with x > x’ > x’’ [57]. For both materials, the rather symmetric shapes suggest a high reversibility of faradaic reactions; however, the higher area of the voltammogram for β3-pH_↗_ indicates a higher capacity. This is confirmed by galvanostatic charge/discharge measurements, illustrated in Figure 7b, which reveal a specific capacity of 17 mAh/g and 32 mAh/g at 2 A/g for β3-pH_14_ and β3-pH_↗_, respectively. In order to better understand the interfacial processes, EIS measurements were performed at an open circuit voltage and the outputs are represented in the Nyquist plot in Figure 7e. The curves are characterized by a semi-circle at high frequencies that is representative of an interfacial charge transfer and by a line at lower frequencies, generally related to the ionic diffusion. The R_CT_ estimated using the equivalent circuit is given in the Appendix A (Appendix A) and reveals a slightly lower charge transfer resistance for β3-pH_↗_ (3.5 Ω for β3-pH_14_ against 2.4 Ω for β3-pH_↗_), as well as a higher slope at low frequencies that highlights better diffusion properties. Thus, these results tend to demonstrate that the presence of numerous small mesopores conjugated to a high SSA (as for β3-pH_↗_) is more favorable for ionic diffusion in the electrode than a porosity essentially composed of big mesopores/macropores and a low SSA as for β3-pH_14_.

The electrode materials synthesized in ionic liquid media present CV profiles similar to β3-pH_↗_ but, as can be observed from their higher area under the CV curves (Figure 7a) and the longer discharge times on the galvanostatic charge–discharge curves (Figure 7b), they possess a higher specific capacity. At 0.2 A/g, the capacities of 40, 42 and 50 mAh/g are measured for β3-pH_↗_(IL 2C), β3-pH_↗_(IL 6C) and β3-pH_↗_(IL 10C), respectively, whereas they are only 19 and 34 mAh/g for β3-pH_14_ and β3-pH_↗_ (Figure 7c). These higher capacities observed for the three β3-pH_↗_(IL) materials could be expected due to the larger specific surface area obtained by nanostructuration, which increases electrode/electrolyte interface and, thus, the number of accessible active redox sites. Moreover, the excellent retention capacity upon high current densities (Figure 7c and Appendix A) and long-term cycling (Figure 7d and Appendix A) already observed for electrode materials without IL are preserved. In fact, all electrode materials possess a capacity retention higher than 80% when increasing the current density from 0.2 to 5 A/g and exhibit almost no capacity loss after 5000 cycles at 2 A/g. Finally, EIS analyses (Figure 7f) also showed that the use of ionic liquid for the nanostructuration reduced the charge transfer resistance (R_CT_ = 0.8, 1.2, 2.2 for β3-pH_↗_(IL 10C), β3-pH_↗_(IL 2C), β3-pH_↗_(IL 6C) respectively against 2.4 for β3-pH_↗_) and improved the transport properties. Although the three cobalt oxyhydroxides synthesized in ionic liquid medium had a comparable specific surface area (Table 2), β3-pH_↗_(IL 10C) possesses the best capacity among them (Figure 7d). This clearly demonstrates that the nature of the ionic liquid used for the synthesis, even when present in small amounts on the surface of the final electrode material, directly affects electrochemical performances. This is especially true when comparing β3-pH_↗_(IL 10C) and β3-pH_↗_(IL 2C); they exhibit exactly the same porosity and the slightly higher SSA for β3-pH_↗_(IL 2C) could logically induce a better capacity, whereas the opposite is experimentally observed. Thus, it seems that the longer the alkyl chain on the imidazolium cycle is, the better the capacity is. The exact reasons for this trend remains to be investigated further, however, this could be connected to the difference in hydrophobic (decyl chain)/hydrophilic (ethyl chain) character between the two ILs, and, thus, the different surface properties. For instance, it can be supposed that the longest alkyl chain facilitates the interfacial proton transfer, as suggested by the lower resistance charge transfer for β3-pH_↗_(IL 10C), leading to better energy storage performance [32]. The ionic liquids have a double beneficial effect on the electrochemical performance; first, used as surfactant/template, they induce nanostructuration that enhances electrode/electrolyte interface and, second, they functionalize the electrode material surface leading to better transport properties and capacities.

Finally, considering the consequent active mass loading of our electrodes, the IL nanostructured cobalt oxyhydroxides exhibit excellent performances (44 mAh/g at 2 A/g and with a mass loading of 5 mg/cm^2^ for β3-pH_↗_(IL 10C)) compared to those reported in the literature. For instance, Wen et al. reported a capacity of 135 F/g (18.8 mAh/g) at 1A/g for mass loading of 2.85 mg/cm^2^, Raj et al. reported 198 F/g (22 mAh/g) at 0.1 A/g for mass loading of 10 mg/cm^2^ and Zhu et al. reported 135 F/g (18.8 mAh/g) and 312 F/g (43 mAh/g) for HCoO_2_ and hybrid HCoO_2_-MWCNT, respectively, at 1 A/g with mass loading around 0.9 mg/cm^2^ [41,43,58].

## 4. Conclusions

This work studies the effects of different synthesis parameters to optimize the nanostructuration of β(III) cobalt oxyhydroxide electrode materials and investigates the effect of this nanostructuration on energy storage performances. First, it is demonstrated that switching from reverse to forward co-precipitation leads to materials with lower crystallinity while maintaining an excellent electronic conductivity, larger specific surface area and thus, better electrochemical performances. Indeed, the high cobalt concentration (supersaturation) in forward co-precipitation favors the nucleation and thus, small crystallites, whereas the reverse co-precipitation in a strong basic medium helps the crystal growth. The use of different imidazolium bromide based ionic liquids, as surfactant/template during the synthesis, further allows to increase the specific surface area and mesoporosity, leading to SSA up to 256 m^2^/g and mesoporosity of 0.43 cm^3^/g for β3-pH_↗_(IL 2C). The length of the alkyl chain (ethyl, hexyl, or decyl) on the 1-alkyl-3-methylimidazolium cation does not have a significant effect on the SSA, but influences the energy storage performance of the electrode materials. Synthesis with IL containing the longest alkyl chain gives the material with the lowest charge transfer resistance and the best capacity in alkaline electrolyte. Finally, by optimizing the nanostructuration, the capacity of high mass loading cobalt oxyhydroxide electrodes could be multiplied by 3, from 17 mAh/g to 43 mAh/g at 2 A/g, while maintaining an excellent capacity retention upon high current density and long-term cycling. As a final point, we believe that this approach of nanostructuration by ILs is very promising and could be successfully extended to other oxide or (oxy)hydroxide electrode materials, such as MnO_2_ or Ni(OH)_2_.

## Figures and Tables

**Figure 1 materials-14-02325-f001:**
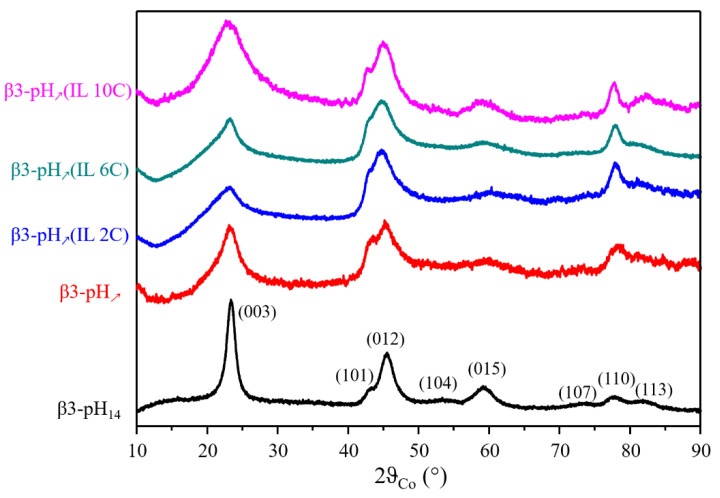
X-ray diffraction patterns of β3-pH_14_ (black), β3-pH_↗_ (red), β3-pH_↗_(IL 2C) (blue), β3-pH_↗_(IL 6C) (green) and β3-pH_↗_(IL 10C) (pink).

**Figure 2 materials-14-02325-f002:**
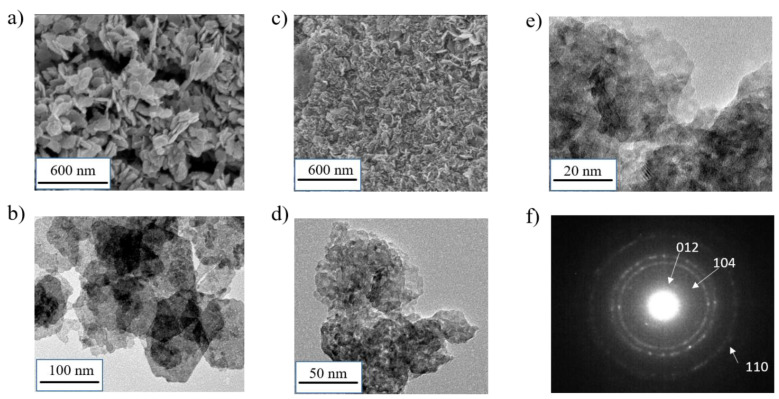
(**a**) SEM images of β3-pH_14_; (**b**) TEM images of β3-pH_14_; (**c**) SEM images of β3-pH_↗_; (**d**,**e**) TEM images of β3-pH_↗_; (**f**) electron diffraction of β3-pH_↗_.

**Figure 3 materials-14-02325-f003:**
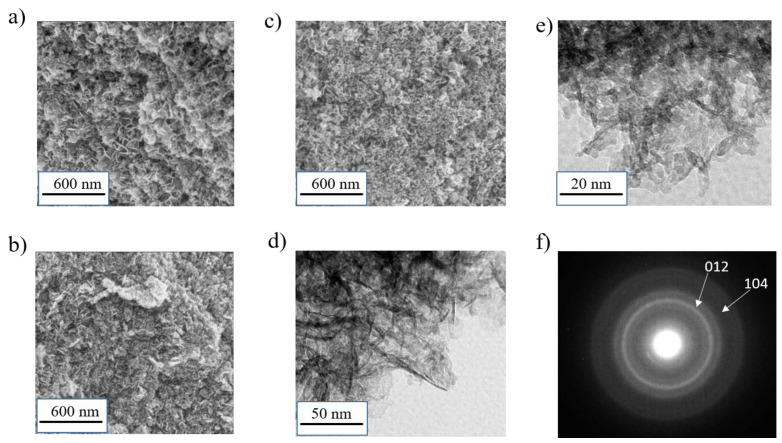
SEM images of (**a**) β3-pH_↗_(IL 2C); (**b**) β3-pH_↗_(IL 6C) and (**c**) β3-pH_↗_(IL 10C); (**d**,**e**) TEM images of β3-pH_↗_(IL 10C); (**f**) electron diffraction of β3-pH_↗_(IL 10C).

**Figure 4 materials-14-02325-f004:**
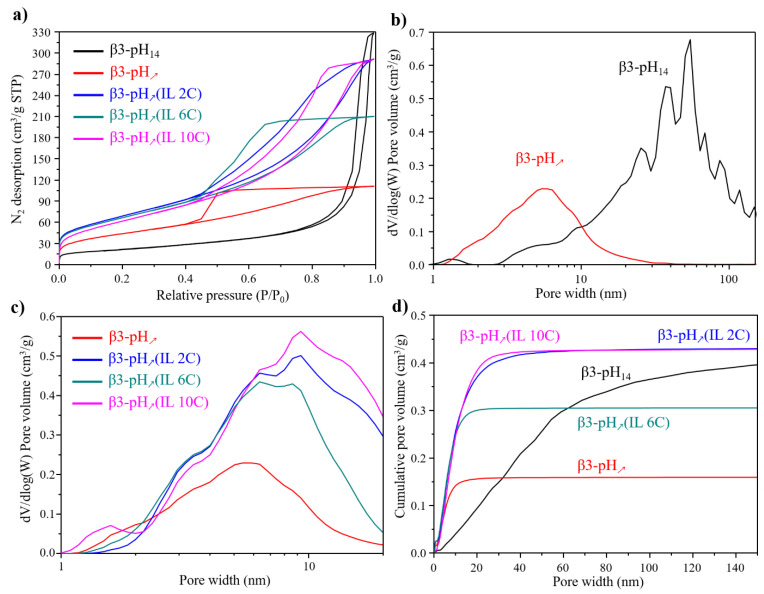
(**a**) N_2_ adsorption/desorption isotherms; (**b**) comparison of pore size distribution between β3-pH_14_ (black) and β3-pH_↗_ (red); (**c**) comparison of pore size distribution between β3-pH_↗_ (red), β3-pH_↗_(IL 2C) (blue), β3-pH_↗_(IL 6C) (green) and β3-pH_↗_(IL 10C) (pink); (**d**) cumulative pore volume depending on the pore width for β3-pH_14_ (black), β3-pH_↗_ (red), β3-pH_↗_(IL 2C) (blue), β3-pH_↗_(IL 6C) (green) and β3-pH_↗_(IL 10C) (pink).

**Figure 5 materials-14-02325-f005:**
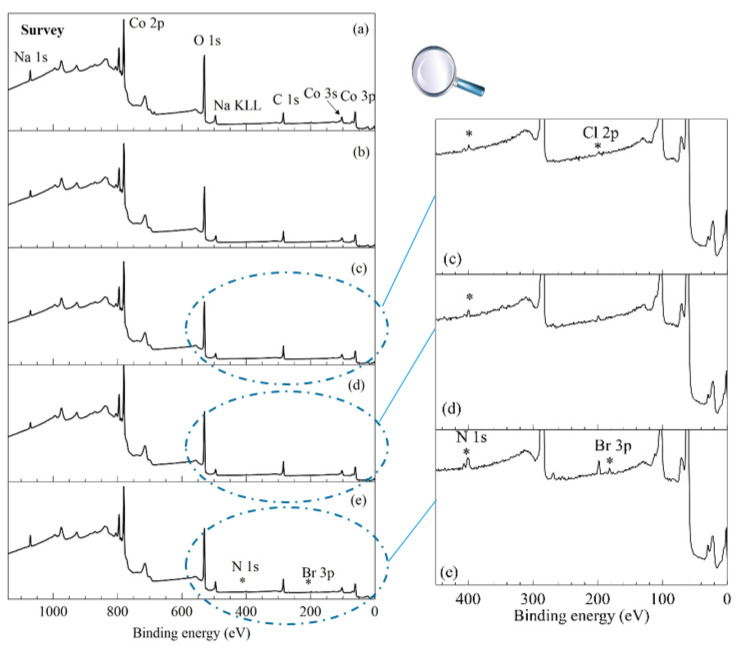
(Left) survey spectra of (**a**) β3-pH_14_, (**b**) β3-pH_↗_, (**c**) β3-pH_↗_(IL 2C), (**d**) β3-pH_↗_(IL 6C) and (**e**) β3-pH_↗_(IL 10C); (right) zoom in of 450-0 eV binding energy range.

**Figure 6 materials-14-02325-f006:**
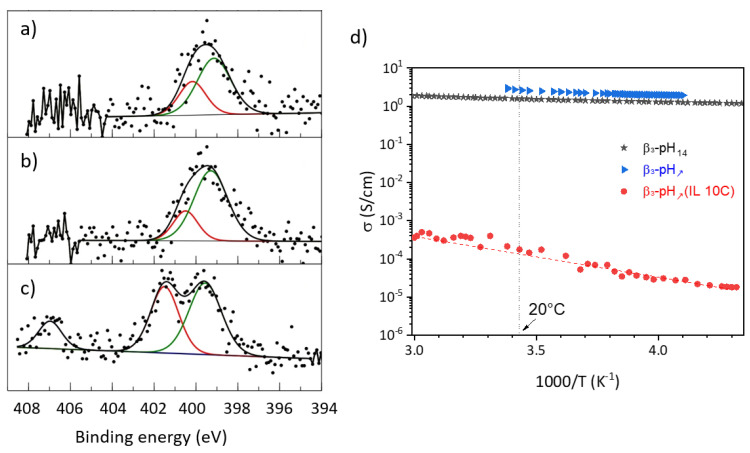
N 1s core peaks spectra (**a**) β3-pH_↗_(IL 2C); (**b**) β3-pH_↗_(IL 6C); (**c**) β3-pH_↗_(IL 10C); (**d**) thermal variation of electronic conductivity for β3-pH_14_ (black pot), β3-pH_↗_ (blue plot) and β3-pH_↗_(IL 10C) (red plot). The samples exhibit room temperature electronic conductivities of 1.8 S·cm^−1^, 2.5 S·cm^−1^ and 2 × 10^−4^ S·cm^−1^ for β3-pH_14_, β3-pH_↗_ and β3-pH_↗_(IL 10C), respectively.

**Figure 7 materials-14-02325-f007:**
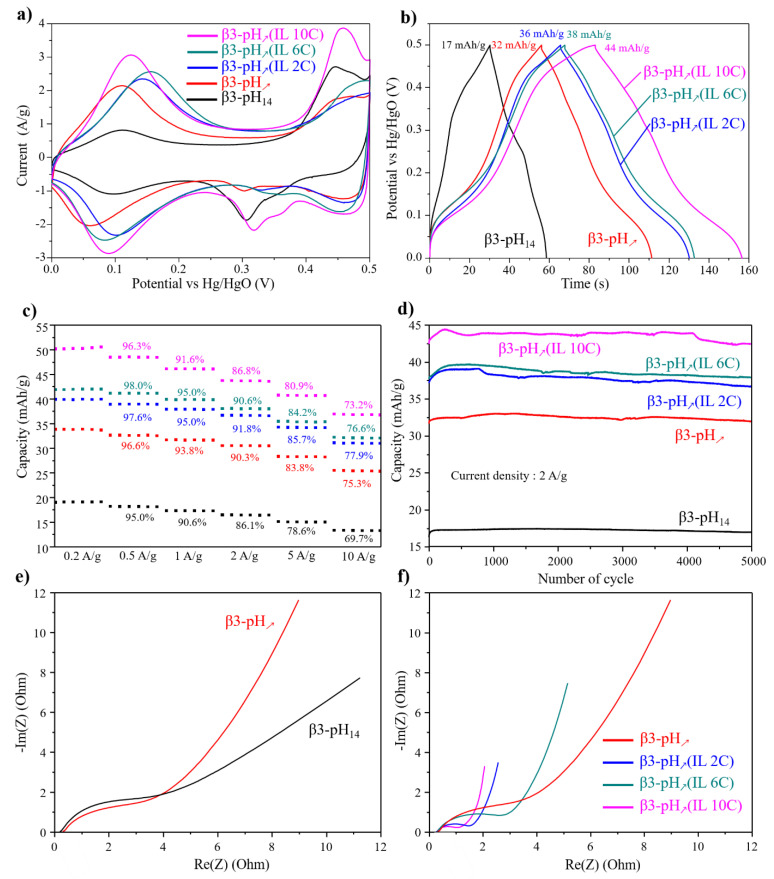
(**a**) Cyclic voltammetry curves measured in 5M-KOH at 5 mV/s for β3-pH_14_ (black curve), β3-pH_↗_ (red curve), β3-pH_↗_(IL 2C) (blue curve), β3-pH_↗_(IL 6C) (green curve) and β3-pH_↗_(IL 10C) (pink curve); (**b**) galvanostatic charge and discharge curves of β3-pH_14_ (black curve), β3-pH_↗_ (red curve), β3-pH_↗_(IL 2C) (blue curve), β3-pH_↗_(IL 6C) (green curve) and β3-pH_↗_(IL 10C) (pink curve) at 2 A/g in 5M-KOH; (**c**) Comparison of capacity retention at different current densities; (**d**) evolution of the capacity upon long-term cycling at 2 A/g; Nyquist plots performed at E_OC_ of (**e**) β3-pH_14_ (black curve), β3-pH_↗_ (red curve) and (**f**) β3-pH_↗_ (red curve), β3-pH_↗_(IL 2C) (blue curve), β3-pH_↗_(IL 6C) (green curve) and β3-pH_↗_(IL 10C) (pink curve).

**Table 1 materials-14-02325-t001:** Size of the coherent domains calculated with the Scherrer equation of the (*003*) and (*110*) reflection peaks, specific surface area and DFT cumulative pore volume for the synthesized cobalt oxyhydroxides.

Sample	Crystallite Size Calculated from (*003*) Reflection Peak (nm)	Crystallite Size Calculated from (*110*) Reflection Peak (nm)	Specific Surface Area (m^2^/g)	DFT Cumulative Pore Volume of Total Porosity and *for pore < 20* nm (cm^3^/g)
β3-pH_14_	10	6	77	0.39–*0.09*
β3-pH_↗_	3	6	160	0.16–*0.16*
β3-pH_↗_(IL 2C)	3	6	256	0.43–*0.37*
β3-pH_↗_(IL 6C)	3	6	246	0.31–*0.31*
β3-pH_↗_(IL 10C)	1.5	6	232	0.43–*0.38*

**Table 2 materials-14-02325-t002:** Chemical composition of different compounds determined by combining chemical analyses and titration method.

Sample	% Na (Weight)	% Co (Weight)	Atomic Ratios	Mean Oxidation State of Co	Chemical Formula
			Na/Co		
β3-pH_14_	1.7	56	0.06	3.15	H_0.79_^+^Na_0.06_^+^(H_2_O)_0.15_Co^3.15+^O_2_
β3-pH_↗_	3.2	54	0.15	3.20	H_0.65_^+^Na_0.15_^+^(H_2_O)_0.5_Co^3.20+^O_2_
β3-pH_↗_(IL 2C)	1.5	45	0.08	3.06	H_0.86_^+^Na_0.08_^+^(H_2_O)_z_Co^3.06+^O_2_(IL)_δ_
β3-pH_↗_(IL 6C)	1.4	42	0.08	3.05	H_0.87_^+^Na_0.08_^+^(H_2_O)_z_Co^3.05+^O_2_(IL)_δ_
β3-pH_↗_(IL 10C)	2.2	44	0.13	3.06	H_0.81_^+^Na_0.13_^+^(H_2_O)_z_Co^3.06+^O_2_(IL)_δ_

## Data Availability

Data are available on demand by asking the corresponding author.

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
