# Peer review of "Controlled Nanostructuration of Cobalt Oxyhydroxide Electrode Material for Hybrid Supercapacitors"

_materials, 2021, doi:10.3390/ma14092325_

Round 1

Reviewer 1 Report

The authors reported a hybrid supercapacitor with the nanostructure β(III) cobalt oxyhydroxides and a serious of ionic liquids. And the effect of the nanostructuration and different ionic liquids on the energy storage performances were studied. Moreover, they found that ILs used as surfactant and template also functionalize the nanomaterial surface, leading to a benefic synergy between the highly ionic conductive IL and the cobalt oxyhydroxide, contributing to the good specific capacity. However, several issues should be addressed before it is considered for publication. 

  1. The statements on XPS data should quote related references. Such as the β3 cobalt oxyhydroxide, and Br 3p3/2 peak.
  2. Please add the equivalentcircuit diagram of EIS curves and quote the related references(10.1002/adma.202005501, etc.).
  3. The galvanostatic charge and discharge curves at different cycles should be supplemented to evaluate the stability.

Author Response

Reviewer 1:

Comments and Suggestions for Authors

The authors reported a hybrid supercapacitor with the nanostructure β(III) cobalt oxyhydroxides and a serious of ionic liquids. And the effect of the nanostructuration and different ionic liquids on the energy storage performances were studied. Moreover, they found that ILs used as surfactant and template also functionalize the nanomaterial surface, leading to a benefic synergy between the highly ionic conductive IL and the cobalt oxyhydroxide, contributing to the good specific capacity. However, several issues should be addressed before it is considered for publication.

Thank you for the review and for the comments

1) The statements on XPS data should quote related references. Such as the β3 cobalt oxyhydroxide, and Br 3p3/2 peak.

We add additional references in the manuscript to attribute the different XPS signals (ref 47 and Ref 48 in the new version)

2) Please add the equivalent circuit diagram of EIS curves and quote the related references (10.1002/adma.202005501, etc.).

The equivalent circuit diagram of EIS was already reported in the supplementary information file (figure S5) and it was mentioned in the electrochemistry part in the text. The reference suggested by the reviewer deals with the challenge of Calcium-ion battery. The review is very interesting, however, we do not think this reference is pertinent in our work.

3) The galvanostatic charge and discharge curves at different cycles should be supplemented to evaluate the stability.

The Galvanostatic charge and discharge curves of the 1st, 3000thand 5000th cycles for all electrode materials were added in the supporting information part. It can be clearly seen that there is only a minor change in capacity retention for all the samples as represented in the figure 7d in the manuscript. The profile of the different curves between the 1st and 5000th cycles remains unchanged which traduces an excellent structural stability. 

Reviewer 2 Report

In this work, the authors fabricated cobalt oxyhydroxide nanomaterials for hybrid supercapacitor electrodes. Ionic liquids were introduced in the preparation processes to control the morphology and the functionality of the materials. The influence of the ionic liquid on the final electrochemical properties of the materials were studied. I suggested the manuscript could be accepted after minor revision.

1. The authors claimed that the fabricated electrodes were battery-type electrodes for hybrid energy storage. These electrodes should obey the relationship i=avb. When b = 1, the electrode process is surface controlled, while b = 0.5 indicates the diffusion-controlled process. [Angew. Chem. Int. Ed. 2021, 60, 5718–5722] Therefore, the reviewer suggests the authors calculate the b values of the electrodes and discuss the corresponding energy storage processes.

2. How did the functionality from the ionic liquids affect the electrochemical performance of the electrodes? The authors should discuss this point in detail.

Author Response

In this work, the authors fabricated cobalt oxyhydroxide nanomaterials for hybrid supercapacitor electrodes. Ionic liquids were introduced in the preparation processes to control the morphology and the functionality of the materials. The influence of the ionic liquid on the final electrochemical properties of the materials were studied. I suggested the manuscript could be accepted after minor revision.

Thank you for the review and for the comments

1) The authors claimed that the fabricated electrodes were battery-type electrodes for hybrid energy storage. These electrodes should obey the relationship i=avb. When b = 1, the electrode process is surface controlled, while b = 0.5 indicates the diffusion-controlled process. [Angew. Chem. Int. Ed. 2021, 60, 5718–5722] Therefore, the reviewer suggests the authors calculate the b values of the electrodes and discuss the corresponding energy storage processes.

Thank you for the comment, indeed we claim that cobalt oxyhydroxide is a battery-type electrode material because it possesses a faradic storage (bulk redox reaction) by comparison to the capacitive storage for Carbon electrode in EDLC devices or pseudocapacitive storage such as in MnO2. The capacitive or pseudocapacitive storage can be clearly distinguish on CV curves and is represented by a rectangular shape. On the other hand, the bulk redox reaction are characterized by redox peak as it is the case for HxCoO2 in this work. Thank you for the reference given by the reviewer, the work is interesting, especially dealing with high mass loading electrodes, and we cited it in the manuscript.

2) How did the functionality from the ionic liquids affect the electrochemical performance of the electrodes? The authors should discuss this point in detail.

This is an excellent question that we try to answer. In this work, we investigate the effect of the alkyl chain length of the ILs and found that the longer alkyl chain, the better energy storage performance are. The ionic liquid is grafted on the cobalt oxyhydroxide and modifies the surface properties. It was reported by Choi et al. (ref 33 ) that this surface modification favors the proton adsorption/desorption that may partially explain the optimization of the energy storage performance between the bare HxCoO2 and the IL functionalized HxCoO2. Moreover, it represents an artificial interface that avoids surface degradation usually leading to capacity fading upon cycling. In this series of electrode materials, we suppose that the length of the alkyl chain which induces a rather hydrophobic/hydrophilic character could influence the performance. We observe the lowest resistance charge transfer for the β3-pH(IL 10C) electrode which means that the longest alkyl chain may favor the interfacial transfer leading to an optimize capacity. A discussion of this point was added in the manuscript.

Reviewer 3 Report

This manuscript described the construction of nanostructuration of β(III) cobalt oxyhydroxide electrode materials by different synthetic method and investigated their affection on the energy storage performances. Especially, they found that the length of the alkyl chain (ethyl, hexyl or decyl) on the 1-alkyl-3-methylimidazolium cation have great influences in the energy storage performances of the electrode materials. This work is well organized, regarding the synthesis and characterization of the different electrodes. This manuscript could be published in Materials after the minor revision.

  1. The authors should modify all the SEM images to a better quality (g., without the original machine’s information).
  2. In Figure 2, the author should provide enlarged SEM photos of (a), (b) to clearly tell readers about the difference between these two materials. Also, it is better for the authors to put a high-resolution TEM images of Figure 2e to show the lattice spacings.
  3. After the stability test, how about the structural change of the β3-pH(IL 10C) electrode?

Author Response

This manuscript described the construction of nanostructuration of β(III) cobalt oxyhydroxide electrode materials by different synthetic method and investigated their affection on the energy storage performances. Especially, they found that the length of the alkyl chain (ethyl, hexyl or decyl) on the 1-alkyl-3-methylimidazolium cation have great influences in the energy storage performances of the electrode materials. This work is well organized, regarding the synthesis and characterization of the different electrodes. This manuscript could be published in Materials after the minor revision.

Thank you for the review and for the positive comments

1) The authors should modify all the SEM images to a better quality (g., without the original machine’s information).

The figures were changed and the original machine’s information were removed

2) In Figure 2, the author should provide enlarged SEM photos of (a), (b) to clearly tell readers about the difference between these two materials. Also, it is better for the authors to put a high-resolution TEM images of Figure 2e to show the lattice spacings.

We add some additional more resolved SEM images in the supplementary information to compare the morphology depending on the synthesis method. On this new images we can clearly seen the platelets like particles for β3-pH14 whereas β3-pH↗ particles are smaller and more wavy. Unfortunately, we do not have the possibility to perform further microscopy imaging these next days to observe the lattice spacing. Although we agree that HR TEM imaging is always a benefit, however, we do not think this experiment would bring further information essential to the manuscript.

3) After the stability test, how about the structural change of the β3-pH(IL 10C) electrode?

Thank you for the interesting question, indeed XRD was done on recovered electrode however due the traces of KOH electrolyte and the initial poor crystallinity of β3-pH(IL 10C), the diagram was mostly amorphous after 5000 cycles. Nevertheless, the Figure S7 that represents the galvanostatic charge and discharge of the 1st, 3000th and 5000th cycles remain very similar which suggests a good structural stability.